

# Major infections following pediatric cardiac surgery pre- and post-CLABSI bundle implementation

Somthida Vachirapuranon[1], Chodchanok Vijarnsorn[1], Supaluck Kanjanauthai[1], Teerapong Tocharoenchok[2], Krivikrom Durongpisitkul[1], Prakul Chanthong[1], Paweena Chungsomprasong[1], Thita Pacharapakornpong[1], Jarupim Soongswang[1], Supattra Rungmaitree[1], Charn Peerananrangsee[3], Ekarat Nitiyarom[2], Kriangkrai Tantiwongkosri[2], Thaworn Subtaweesin[2] and Amornrat Phachiyanukul[4]

[1] Department of Pediatrics, Faculty of Medicine Siriraj Hospital, Mahidol University, Bangkok, Thailand
[2] Department of Surgery, Faculty of Medicine Siriraj Hospital, Mahidol University, Bangkok, Thailand
[3] Department of Pharmacology, Faculty of Medicine Siriraj Hospital, Mahidol University, Bangkok, Thailand
[4] Pediatric Nursing Division, Department of Nursing, Faculty of Medicine Siriraj Hospital, Mahidol University, Bangkok, Thailand

Corresponding author
Chodchanok Vijarnsorn, cvijarnsorn@yahoo.com

## ABSTRACT

**Background.** Postoperative infection contributes to the worsening of congenital cardiac surgery (CCS) outcomes. Surgical site infection (SSI), bloodstream infection (BSI) and ventilator associated pneumonia (VAP) are common. An additional bundle of preventive measures against central-line associated bloodstream infection (CLABSI) bundle was implemented in April 2019.

**Objectives.** To compare the incidence of major infections after pediatric CCS before and after the implementation of the CLABSI bundle and to identify risk factors for major infections.

**Methods.** We conducted a single-center, retrospective study to assess the incidence of major infections including bloodstream infection (BSI), surgical site infection (SSI), and ventilator-associated pneumonia (VAP) after pediatric CCS one year before and after implementation of the CLABSI bundle during April 2018–March 2020. The demographics and outcomes of the patients were explored, and risk factors for major infections were identified using multivariate analysis.

**Results.** A total of 548 children (53% male) underwent CCS with a median age of 1.9 years (range 0.01–17.5 years). The median Aristotle Basic Complexity score was 7.1 (range 3–14.5). The CLABSI bundle was applied in 262 patients. Overall mortality was 5.5%. 126 patients (23%) experienced major postoperative infections. During the year after the implementation of the CLABSI bundle, BSI was reduced from 8.4% to 3.1% ($p = 0.01$), with a smaller reduction in VAP (21% to 17.6%; $p = 0.33$). The incidence of SSI was unchanged (1.7% to 1.9%; $p = 0.77$). The independent risk factors for major infections were age at surgery <6 months ($p = 0.04$), postoperative ventilator usage >2 days ($p < 0.01$), central line usage >4 days ($p = 0.04$), and surgery during the pre-CLABSI bundle period ($p = 0.01$).

**Conclusion.** Following the implementation of the CLABSI prevention package in our pediatric CCS unit, the incidence of BSI was significantly reduced. The incidence of

VAP tended to decrease, while the SSI was unchanged. Sustainability of the prevention package through nurse empowerment and compliance audits is an ongoing challenge.

# INTRODUCTION

Congenital cardiac surgery (CCS) has become more common in developing countries and outcomes have improved. Children with CCS are at high risk for healthcare–associated infections (HAIs), leading to considerable morbidity and mortality, increased use of antibiotics, and longer hospital stays. Surgical site infection (SSI), bloodstream infection (BSI), and ventilator-associated pneumonia (VAP) are common after cardiac surgery (*Alten et al., 2018*; *Axelrod et al., 2015*; *Barker et al., 2010*; *Sen et al., 2017*). The incidence of major infections in pediatric CCS has been reported to be between 15% and 30% (*Barker et al., 2010*; *Sen et al., 2017*; *Levy et al., 2003*; *Vijarnsorn et al., 2012*; *Chang et al., 2020*). Risk factors for postoperative infections following CCS include younger age at surgery and complexity of the procedure (*Alten et al., 2018*; *Axelrod et al., 2015*; *Barker et al., 2010*; *Sen et al., 2017*; *Levy et al., 2003*). In-hospital mortality, prolonged duration of mechanical ventilation, and long stays in the intensive care unit (ICU) have been well-documented (*Sen et al., 2017*; *Vijarnsorn et al., 2012*; *Agarwal et al., 2014*). To improve safety and quality for children undergoing congenital heart surgery in developing countries, the International Quality Improvement Collaborative (IQIC) developed and piloted a safe surgical checklist, team-based practice, and a focus on hand hygiene (*Sen et al., 2017*; *Jenkins et al., 2014*; *Balachandran et al., 2015*). The primary aim was to reduce mortality associated with CCS, but also resulted in a considerable reduction in bacterial sepsis and SSI. For example, a study by Balachandran et al. reported a significant reduction in bacterial sepsis from 15.1% in 2010 to 9.6% in 2012 after launching the IQIC program (*Balachandran et al., 2015*).

Our pediatric cardiac service has seen an increase in surgical demand and complexity. In 2005, the mortality following CCS in our center was 6.1%. We identified high complexity, bypass time >85 min, and cross clamp time >60 min as the main risk factors for mortality (*Vijarnsorn et al., 2011*). At the time, 13.5% of our patients experienced major infections such as BSI, SSI, VAP, or urinary tract infection (*Vijarnsorn et al., 2012*). In 2013, we implemented a safe surgical checklist and a hand hygiene program as described in the IQIC campaign. In 2016, we introduced a VAP prevention package consisting of strategies for weaning, hygienic hand washing, aspiration precaution, prevention of contamination, and chest physiotherapy; together known as the WHAP-C bundle (*Smulders, Van Gestel & Bos, 2013*; *Chang & Schibler, 2016*). Use of checklists, regular audits and reassessments are common practices to ensure adherence to the bundles.

In our center between 2018 and early 2019, central-line associated bloodstream infections (CLABSI) and VAP with antibiotic resistant bacteria such as carbapenem-resistant enterobacteriacae (CRE) were identified following CCS in pediatric cardiac care units

(PCCU). These infections required extensive contact precautions and prolonged antibiotic treatments (*Guh, Limbago & Kallen, 2014*). There is growing evidence that the addition of a maintenance bundle to a central-line insertion bundle can reduce the incidence of CLABSI in critically ill patients of all ages (*Smulders, Van Gestel & Bos, 2013*; *Ista et al., 2016*; *Hussain et al., 2017*; *Costello et al., 2008*). Beginning in April 2019, our institutional infection control committee and the PCCU team started a modified pediatric CLABSI bundle for all children undergoing CCS. The principal components of the CLABSI bundle (*Ista et al., 2016*; *Hussain et al., 2017*) are illustrated in Fig. 1. We modified additional practices to prevent CRE from the report by *Guh, Limbago & Kallen (2014)* (Fig. 2) and used 80% alcohol spray to disinfect the three-way extension tube that connects to the central line (*Oishi et al., 2004*). The modified CLABSI bundle was implemented using a team-based approach. Following one month of training the prevention package to all medical staff and nurses in the unit, infection control personnel provided detailed education to physicians and nurses, daily goal sheets were established for timely removal of the central venous line, and real-time feedback on the CLABSI data was used to monitor the program. Processes and procedures were often observed directly by unit and infection control personnel, and team discussions were held regularly to identify obstacles and track progress (*Chang et al., 2020*; *Smulders, Van Gestel & Bos, 2013*; *Hussain et al., 2017*; *Costello et al., 2008*). Physicians and nurses were encouraged to assist each other to be aware of and adhere to the maintenance bundles.

Therefore, we aimed to measure the effect of the CLABSI bundle on the incidence of major infections such as BSI, SSI, and VAP after pediatric CCS. We hypothesized that implementation of the CLABSI bundle would reduce BSI, VAP and SSI. We also explored risk factors for the occurrence of major infections in children undergoing CCS at Siriraj Hospital between 1 April 2018 and 31 March 2020.

## MATERIALS AND METHODS

This retrospective study was approved by the Siriraj Institutional Review Board, Faculty of Medicine, Siriraj Hospital, Mahidol University (Study number 258/2563(IRB4), COA Si 275/2020). Informed consent from patients was waived and multiple steps were taken to ensure the protection of patient confidentiality. All research methods were performed according to Good Clinical Practice (GCP) guidelines and regulations.

Data were reviewed from children <18 years old who underwent CCS at Siriraj hospital during a one-year period pre-CLABSI bundle (1 April 2018–31 March 2019), and one year post-CLABSI bundle implementation (1 April 2019–31 March 2020). Preterm infants (gestational age <37 weeks), patients with underlying disease such as autoimmune disease or cancer, and patients with extracardiac surgery in the same operation such as tracheoplasty, gastrostomy were excluded from the study. Demographic data, including age, gender, weight, height, diagnosis of congenital heart disease, genetic syndromes, preoperative history of major infection within three months prior to operation, preoperative ventilator use, central line usage, presence of clinical heart failure/pulmonary hypertension, previous cardiovascular surgery and intraoperative data, including type of operation and complexity

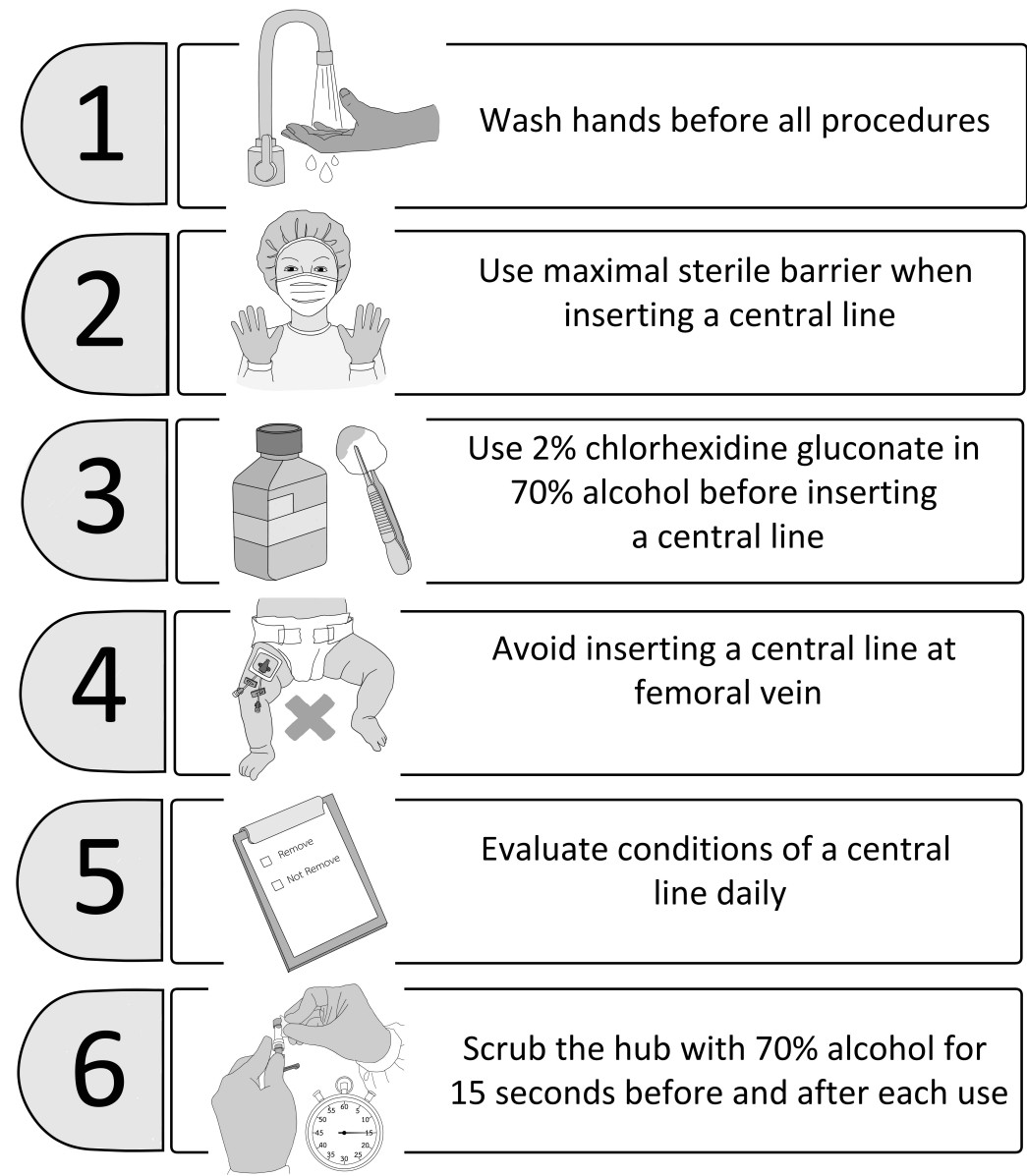

**Figure 1** **The CLABSI bundle used in our pediatric cardiac care units.** CLABSI, central-line associated bloodstream infection.

of the surgical procedure using the Aristotle Basic Complexity score (ABC score), operative time, cardiopulmonary bypass time (CPB time), aortic cross clamp time (AoX time), delayed sternal closure, and extracorporeal membrane oxygenator (ECMO) were collected.

Postoperative outcomes included in-hospital mortality, duration of ventilator use, hospital length of stay (LOS) and major infections; BSI, SSI and VAP. BSI was defined as any two clinical signs of sepsis (fever or hypothermia, tachycardia, hypotension, tachypnea, leukocytosis or leucopenia) with a positive blood culture (*Chang et al., 2020*; *Haughey, White & Seckeler, 2019*). SSI was defined as an infection related to an incisional procedure

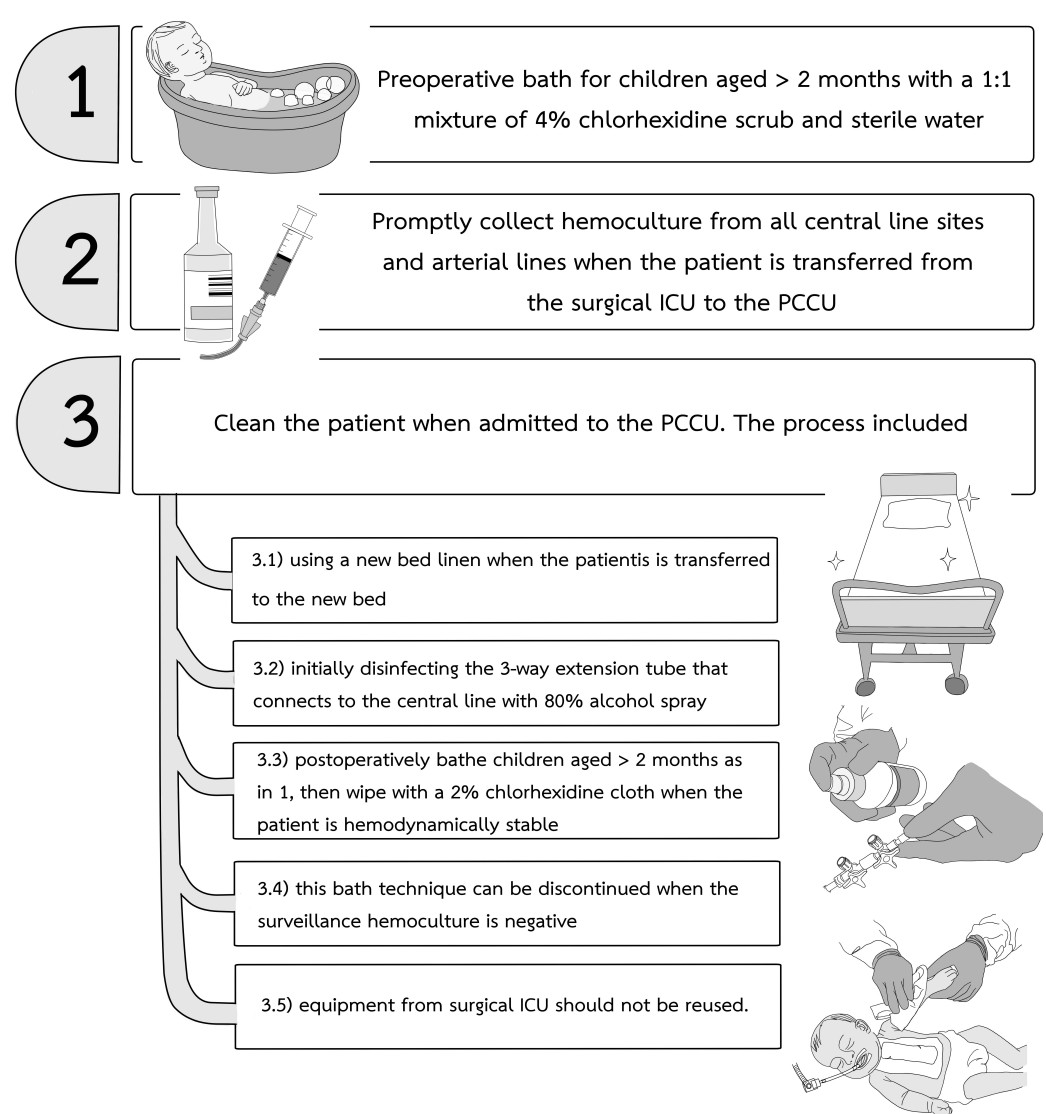

**Figure 2** The CRE prevention protocol used together with the CLABSI bundle in our pediatric cardiac care units. CRE, carbapenem-resistant enterobacteriacae; CLABSI, central-line associated bloodstream infection.

that included superficial incisional SSI, deep incisional SSI or mediastinitis (*Chang et al., 2020*; *Hodge et al., 2019*). VAP was defined as a pneumonia in patients on a mechanical ventilator for >2 days with evidence of new pulmonary infiltration, signs of systemic infection or detection of causative agents (*Chang & Schibler, 2016*). The outcomes were compared between pre- and post-CLABSI bundle implementation.

## Statistical methods
Statistical analyses were performed using SPSS 20.0 for Windows (SPSS Inc., Chicago, IL, USA). Demographic data, preoperative and perioperative data were presented as a frequency with percentages for categorical variables and mean ± SD or median with range

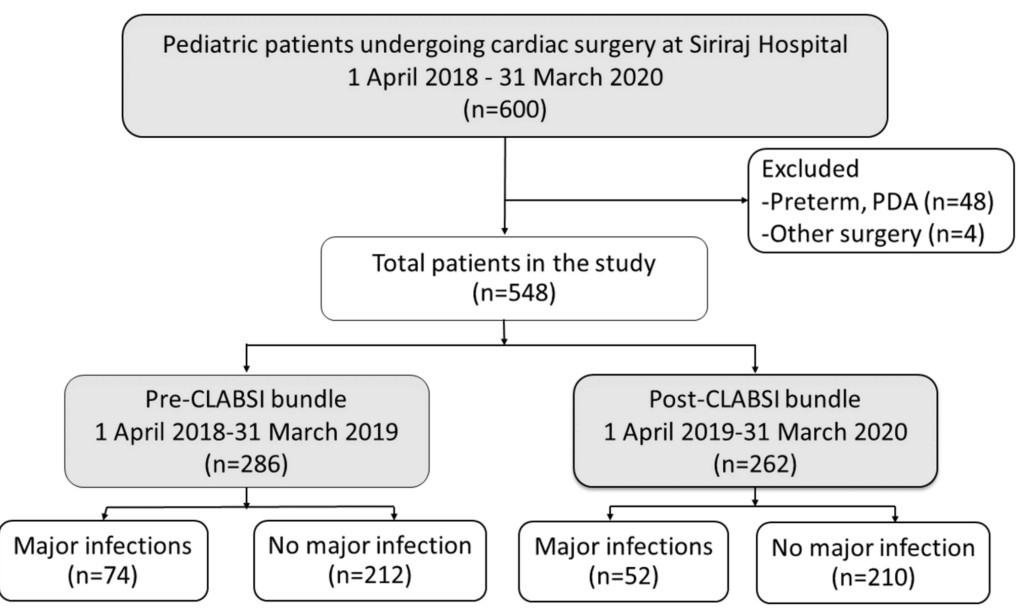

**Figure 3** **Flow diagram of pediatric patients included in the analysis (**$n = 548$**).** CLABSI, central-line associated bloodstream infection; PDA, patent ductus arteriosus.

for continuous variables. Mortality and major infections between pre- and post-CLABSI bundle implementation were analyzed and compared using chi-square. Factors associated with major infection following CCS and mortality were analyzed using univariate and multivariate analyses with logistic regression. A $p$-value <0.05 was considered statistically significant.

## RESULTS

### Patient characteristics
Five hundred and forty-eight pediatric patients were included in the analysis (Fig. 3). The demographic, clinical, and operative management of the patients are shown in Table 1. The patients had a median age of 1.9 years (0.01–17.5 years). Two hundred and ninety-three patients (53%) were male, and 96 patients (12.6%) had a genetic syndrome. The median Aristotle Basic Complexity score was 7.1 (range 3–14.5). Four hundred and ninety-two patients (89%) underwent open-heart surgery.

The overall in-hospital mortality of pediatric CCS was 5.5% ($n = 30$) (Table 2). Major infections (BSI, SSI and VAP) were reported in 126 patients (23%) (Fig. 4). Twenty-two patients had two major infections in their postoperative course. Among the 286 patients in the pre-CLABSI bundle period, 24 patients (8.4%) had BSI, while eight of the 262 patients (3.1%) in the post-CLABSI bundle period had BSI ($p = 0.01$). The pathogens causing BSI included Methicillin-Resistant Coagulase-Negative Staphylococci ($n = 11$), *Klebsiella pneumonia* e ($n = 6$), *Acinetobacter baumannii* ($n = 6$), *Stenotrophomonas maltophilia* ($n = 4$), *Candida parapsilosis* ($n = 2$), *Staphylococcus haemolyticus* ($n = 1$), *Alpha-hemolyticus streptococci* ($n = 1$), and *Escherichia coli* ($n = 1$). SSI occurred in 1.7%

**Table 1  Baseline characteristics.**

| | Total ($n =548$) | Pre-CLABSI bundle ($n =286$) | Post-CLABSI bundle ($n =262$) | *p*-value |
|---|---|---|---|---|
| Age at diagnosis at Siriraj Hospital (years) | 0.51 (0, 15.47) | 0.53 (0, 15.47) | 0.46 (0, 14.6) | 0.484 |
| Age at surgery (years) | 1.96 (0.01, 17.40) | 2.11 (0.01, 17.40) | 1.82 (0.01, 15.04) | 0.347 |
| Gender | | | | 0.484 |
| Male | 293 (53.5%) | 157 (54.9%) | 136 (51.9%) | |
| Female | 255 (46.5%) | 129 (45.1%) | 126 (51.9%) | |
| Weight (kg) | 12.56 ± 11.05 | 12.66 ± 10.61 | 12.45 ± 11.53 | 0.829 |
| Height (cm) | 86.92 ± 29.12 | 88.97 ± 28.64 | 84.68 ± 29.52 | 0.092 |
| Diagnosis | | | | 0.124 |
| Cyanotic heart disease | 295 (53.8%) | 145 (50.7%) | 150 (57.3%) | |
| Acyanotic heart disease | 253 (46.2%) | 141 (49.3%) | 112 (42.7%) | |
| Presence of genetic syndrome | 69 (12.6%) | 39 (13.6%) | 30 (11.5%) | 0.441 |
| Down syndrome | 37 (6.8%) | 22 (7.7%) | 15 (5.7%) | 0.359 |
| DiGeorge syndrome | 12 (2.2%) | 5 (1.7%) | 7 (2.7%) | 0.461 |
| CHARGE association | 2 (0.4%) | 0 (0.0%) | 2 (0.8%) | 0.228 |
| other | 18 (3.3%) | 12 (4.2%) | 6 (2.3%) | 0.238 |
| Asplenia or polysplenia | 42 (7.7%) | 19 (6.6%) | 23 (8.8%) | 0.348 |
| Presence of heart failure | 261 (47.6%) | 150 (52.4%) | 111 (42.4%) | 0.018[*] |
| Presence of pulmonary hypertension | 66 (12%) | 36 (12.6%) | 30 (11.5%) | 0.683 |
| Previous of cardiovascular surgery | 121 (22.1%) | 65 (22.7%) | 56 (21.4%) | 0.703 |
| History of major infection within 3 months prior to operation | 79 (14.4%) | 43 (15%) | 36 (13.7%) | 0.667 |
| Preoperative functional class | | | | <0.001[*] |
| Functional class I | 60 (10.9%) | 17 (5.9%) | 43 (16.4%) | |
| Functional class II | 274 (50%) | 172 (60.1%) | 102 (38.9%) | |
| Functional class III | 181 (33%) | 84 (29.4%) | 97 (37%) | |
| Functional class IV | 33 (6%) | 13 (4.5%) | 20 (7.6%) | |
| Preoperative usage of ventilator | 32 (5.8%) | 18 (6.3%) | 14 (5.3%) | 0.636 |
| Preoperative usage of central line | 56 (10.2%) | 30 (10.5%) | 26 (10.0%) | 0.839 |
| Preoperative antibiotics beyond usual prophylaxis usage | 46 (8.4%) | 24 (8.4%) | 22 (8.4%) | 0.998 |
| Procedure Aristotle Basic Complexity score | 7.10 ± 2.02 | 7.02 ± 1.99 | 7.20 ± 2.06 | 0.266 |
| Procedure related with systemic to pulmonary shunt | 63 (11.5%) | 36 (12.7%) | 27 (10.3%) | 0.378 |
| Emergency or urgency procedure | 22 (4%) | 20 (7%) | 2 (0.8%) | <0.001[*] |
| Open heart surgery | 492 (89.8%) | 266 (93%) | 226 (86.3%) | 0.009[*] |
| Operative time (min) | 184.93 ± 93.28 | 179.62 ± 87.09 | 190.74 ± 99.45 | 0.163 |
| Cardiopulmonary time (min) | 106.96 ± 56.93 | 100.83 ± 51.04 | 113.40 ± 61.99 | 0.018[*] |
| Aortic cross clamp time (min) | 68.58 ± 53.04 | 63.62 ± 37.28 | 73.93 ± 65.66 | 0.053 |
| Perioperative and postoperative ECMO | 23 (4.2%) | 12 (4.2%) | 11 (4.2%) | 0.999 |
| Delayed sternal closure | 32 (5.8%) | 14 (4.9%) | 18 (6.9%) | 0.325 |

**Notes.**

Data presented as n (%), mean ± SD and median (range).

*Statistical significance at *p*-value <0.05.

CLABSI, central-line associated blood stream infection; CHARGE, coloboma, heart defects, atresia choanae, growth retardation, genital abnormalities and ear abnormalities; min, minutes; ECMO, extracorporeal membrane oxygenator.

**Table 2  Postoperative outcomes and complications.**

|  | Total ($n=548$) | Pre-CLABSI bundle ($n=286$) | Post-CLABSI bundle ($n=262$) | *p*-value |
|---|---|---|---|---|
| Requirement of RRT | 21 (3.8%) | 12 (4.2%) | 9 (3.4%) | 0.664 |
| Presence of multi-organ dysfunction | 54 (9.9%) | 24 (8.4%) | 30 (11.5%) | 0.253 |
| Stroke | 2 (0.4%) | 2 (0.7%) | 0 (0.0%) | 0.500 |
| Seizure required AED | 16 (2.9%) | 8 (2.8%) | 8 (3.1%) | 1.000 |
| Bed ridden | 2 (0.4%) | 1 (0.3%) | 1 (0.4%) | 1.000 |
| Loss limb (gangrene) | 3 (0.5%) | 1 (0.3%) | 2 (0.8%) | 0.609 |
| Hospital length of stays (day) | 14.41 ± 19.17 | 13.76 ± 17.93 | 15.11 ± 20.45 | 0.411 |
| Ventilator usage days | 4.55 ± 10.92 | 4.45 ± 11.87 | 4.65 ± 9.79 | 0.834 |
| Central line usage days | 8.94 ± 11.56 | 8.74 ± 11.75 | 9.16 ± 11.38 | 0.669 |
| Postoperative functional class III-IV | 44 (8.0%) | 17 (5.9%) | 27 (10.3%) | 0.083 |
| Major infection (BSI, SSI, VAP) | 126 (22.9%) | 74 (31.1%) | 52 (22.5%) | 0.09 |
| Mortality | 30 (5.5%) | 16 (5.6%) | 14 (5.3%) | 1.000 |

**Notes.**

Data presented as n (%), mean ± SD and median (range).

*Statistical significance at *p*-value <0.05.

CLABSI, central-line associated blood stream infection; RRT, renal replacement therapy; AED, antiepileptic drug; BSI, bloodstream infection; SSI, surgical site infection; VAP, ventilator-associated pneumonia.

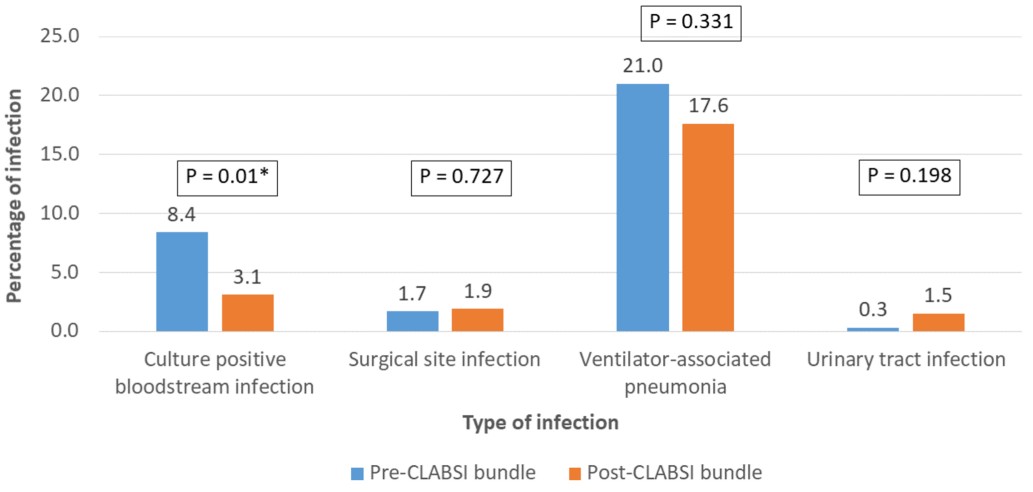

**Figure 4  Comparison of the incidence of major infections; bloodstream infection/culture positive septicemia, surgical site infection, and ventilator-associated pneumonia before the central-line associated bloodstream infection (CLABSI) bundle ($n = 286$) and after the implementation of the CLABSI bundle ($n = 262$).**

and 1.9% of patients in the pre- and post-bundle implementation period, respectively ($p = 0.727$). There was a reduction in the incidence of VAP after the implementation of the CLABSI bundle (21% to 17.6%), however; the incidence of VAP did not decrease significantly ($p = 0.331$).

**Table 3  Risk factors for major postoperative infection (n = 548).**

| Variables | Crude OR (95% CI) | p-value | Adjusted OR (95% CI) | p-value |
|---|---|---|---|---|
| Male gender | 0.825 (0.552, 1.232) | 0.346 | | |
| Age at surgery <6 months | 5.025 (3.225, 7.812) | <0.001* | 1.870 (1.019, 3.432) | 0.043* |
| Weight <5 kg | 4.347 (2.724, 6.535) | <0.001* | | |
| Presence of genetic syndrome | 1.110 (0.617, 1.997) | 0.728 | | |
| Single ventricle | 1.850 (1.100, 3.112) | 0.019* | 1.902 (0.869, 4.160) | 0.108 |
| Asplenia or polysplenia | 1.759 (0.896, 3.455) | 0.097 | 0.781 (0.283, 2.156) | 0.633 |
| History of major infection within 3 months prior to operation | 3.304 (2.005, 5.445) | <0.001* | 1.162 (0.589, 2.294) | 0.665 |
| Preoperative functional class III-IV | 2.745 (1.826, 4.126) | <0.001* | 1.408 (0.841, 2.358) | 0.193 |
| Preoperative usage of ventilator | 4.857 (2.341, 10.078) | <0.001* | 0.805 (0.299, 2.165) | 0.668 |
| Preoperative usage of central line | 5.182 (2.923, 9.185) | <0.001* | 1.413 (0.635, 3.142) | 0.397 |
| Procedure Aristotle Basic Complexity score >9 | 2.021 (1.137, 3.592) | 0.015* | 0.774 (0.346, 1.731) | 0.553 |
| Type of surgery: emergency or urgency | 3.574 (1.511, 8.454) | 0.002* | 1.844 (0.607, 5.604) | 0.281 |
| Operative time >240 min | 1.764 (1.125, 2.766) | 0.013* | 1.244 (0.661, 2.341) | 0.499 |
| CPB time >90 min | 1.409 (0.945, 2.101) | 0.092 | 1.036 (0.596, 1.803) | 0.901 |
| Delayed sternal closure | 4.232 (2.048, 8.744) | <0.001* | 0.995 (0.397, 2.494) | 0.992 |
| Ventilator usage >2 days | 7.671 (4.897, 12.030) | <0.001* | 3.942 (2.222, 6.997) | <0.001* |
| Central line usage >4 days | 6.993 (3.533, 13.604) | <0.001* | 2.240 (1.017, 4.932) | 0.045* |
| Pre CLABSI bundle period | 1.410 (0.990, 2.109) | 0.094 | 1.827 (1.128, 2.959) | 0.014* |

**Notes.**
Multivariate analysis by logistic regression.
*Statistical significance at p-value <0.05.
OR, Odd ratio; min, minutes; CPB, cardiopulmonary bypass; CLABSI, central-line associated bloodstream infection.

## CONSEQUENCES AND RISKS OF MAJOR INFECTION

In children with major postoperative infection, hospital LOS and ventilator use were significantly higher (median LOS 17.5 days vs 7 days; $p < 0.001$; median ventilator use 5 days vs 1 day; $p < 0.001$). Mortality was 11.1% in patients with major infection compared to 3.8% in those without infection ($p = 0.002$). The proportion of patients with/without major infections are shown with risk variables in Table S1. The adjusted OR for major postoperative infections in the pre-CLABSI bundle period was 1.83 (95% CI [1.13–2.96]; $p = 0.01$). Independent risk factors for major postoperative infections were age at surgery less than 6 months (adjusted OR 1.871; 95% CI [1.02–3.43]; $p = 0.04$), ventilator usage >2 days (adjusted OR 3.94; 95% CI [2.22–6.99]; $p < 0.001$) and central line usage >4 days (adjusted OR 2.20; 95% CI [1.02–4.93]; $p = 0.04$) (Table 3). The risk factors for mortality are presented in Table S2. Factors that independently increased the mortality risks were single ventricle (adjusted OR 6.66; 95% CI [1.76–25.02]; $p$-value=0.005), emergent or urgent procedure (adjusted OR 5.35; 95% CI [1.24–22.98]; $p$-value=0.02), and operative time >240 min (adjusted OR 3.79; 95% CI [1.19–12.01]; $p$-value = 0.02).

## DISCUSSION

Using a pre- and post-intervention design, we measured significant reductions in BSI from 8.4% to 3.1% ($p = 0.01$). The incidence of VAP did not decrease significantly (21% to

17.6%; $p = 0.33$), and the SSI rate remained unchanged at less than 2%. A team-based approach, continuous education, real-time feedback, and direct observation monitoring were key to successful implementation. Independent risk factors for major postoperative infections were age at surgery less than six months, postoperative ventilator usage >2 days, central line usage >4 days, and pre-CLABSI bundle implementation period.

We observed that hospital LOS and ventilator use in children with major postoperative infection were much higher ($p < 0.001$), underscoring the importance of preventing these infections in this vulnerable population. Following an outbreak of antibiotic- resistant bacterial infection in our PCCU, we modified the standard CLABSI bundle for children and neonates, with specific changes based on unit-based consensus and consultation with infection control experts (*Smulders, Van Gestel & Bos, 2013*; *Ista et al., 2016*; *Hussain et al., 2017*). An alcohol spray of 80% was used to disinfect three-way ports and extension tubes that connect to the central line as soon as the patient was transferred to the PCCU bed and, to prevent CRE, pre- and postoperative baths were administered to children older than 2 months using 4% chlorhexidine scrub diluted 1:1 in sterile water (*Guh, Limbago & Kallen, 2014*; *Oishi et al., 2004*). No skin reaction was reported after preoperative and postoperative baths in the study. It is underlined that standard central venous catheter cares including catheter flushing, dressing, blood withdrawal, heparin flushing to thrombosis prevention were also emphasized and performed with aseptic technique as neonate or pediatric protocol along with modified CLABSI bundle implementation. Maintenance and sustained application of the prevention bundle using team-based monitoring and performing constructive compliance audits is an ongoing process.

## BSI and SSI

We identified 42 patients (7.6%) with BSI or SSI, and one patient had both BSI and SSI. During the two-year period, the incidence of BSI (5.8%) and SSI (1.8%) in our center was consistent with the IQIC data published in 2017 of the developing world congenital heart surgery programs (*Sen et al., 2017*). We observed a decrease in the incidence of our patients with BSI from 8.4% to 3.1% after the implementation of the CLABSI bundle. Nonetheless, this was higher than the 2019 data from a university hospital in the US where just 1.5% of neonates and 0.8% of infants experienced a BSI (*Haughey, White & Seckeler, 2019*).

## VAP

The overall incidence of VAP following pediatric CCS during the two-year period was high (19.3%). The definition of VAP in children varies and is somewhat controversial. A recent, age-adjusted definition of VAP requires mechanical ventilation (generally >2 days), clinical signs of infection, change in sputum volume, worsening gas exchange and radiological criteria (*Chang & Schibler, 2016*). A positive microbiological culture of a tracheal aspirate or bronchoalveolar lavage specimen is preferable, but not essential for the diagnosis of VAP (*Chang & Schibler, 2016*; *Vijay et al., 2018*). Therefore, some of our patients with VAP had culture-negative pneumonia, and this may have resulted in an overestimation of the number of VAP patients when compared with adult criteria. The incidence of VAP in global pediatric intensive care units varies from 2% to 38.4%, with a higher incidence in

developing countries (*Chang & Schibler, 2016*; *Vijay et al., 2018*). The incidence of VAP following pediatric CCS in our study was higher than studies from Taiwan (13%) (*Tang et al., 2009*) and the Netherland (8.8%). Although the application of the VAP bundle plus the CLABSI bundle tended to reduce our rate from 21% to 17.6% ($p = 0.33$), we are not satisfied and intend to redouble our efforts to improve the quality of respiratory care in pediatric CCS.

### Risk factors of major infections

Younger age at surgery, postoperative ventilator usage >2 days, central line usage >4 days, and pre-CLABSI bundle period were independent predictors of major infections. The large database of developing countries from the IQIC showed that independent risks of BSI and SSI were younger age at surgery, greater surgical complexity, lower oxygen saturation, and major medical illness (*Sen et al., 2017*). *Barker et al. (2010)* examined major infections including septicemia, mediastinitis, and endocarditis using the Society of Thoracic Surgeons Congenital Heart Surgery Database and identified that young age, high complexity, previous cardiothoracic operation, preoperative stay more than one day, preoperative ventilator support, and presence of a genetic abnormality were significant risks of infection in a multivariate model. In 2005, we identified risk factors for BSI, SSI, VAP, and UTI were infancy, prolonged ventilator support >2 days, hospital LOS >14 days, intensive care unit (ICU) LOS >3 days, re-opening surgery and extubation failure rate (*Vijarnsorn et al., 2012*).

### Study limitations

Our study has several limitations. Our subjects were not randomized and our analysis was retrospective in nature, a study design that leaves open the possibility of bias, known and unknown. For example, BSI requiring antibiotics may be confused with systemic post-CPB inflammatory response (SIR). Therefore, BSI in the study was counted only in patients with at least two clinical signs of sepsis and a positive blood culture. Conversely, VAP was included culture negative pneumonia. This may have led us to under or overestimate the incidence of BSI and VAP. Notably, our analysis included only a single year pre- and post-CLABSI bundle implementation; it is likely that more data will yield additional insights. Finally, we did not examine the medium or long-term outcomes of our patients, nor did we explore the financial costs and benefits of the CLABSI bundle. These considerations deserve our attention in future studies.

## CONCLUSION

Following the implementation of the CLABSI prevention bundle in our pediatric CCS unit, the incidence of BSI was reduced significantly. The incidence of VAP tended to decrease, while the SSI was unchanged. Sustained application of the prevention bundle using team-based monitoring, empowering nurses to prevent infections, and performing constructive compliance audits is an ongoing challenge.

## ACKNOWLEDGEMENTS

The authors acknowledge the medical faculty and nurses of Cardiovascular Thoracic Surgery and Pediatrics, Faculty of Medicine Siriraj Hospital, for their support and involvement in patient care. The authors also thank Miss Kanokwan Sommai and Dr. Julaporn Pooliam, Clinical Epidemiology Unit, Office of Research and Development, Faculty of Medicine, Siriaj Hospital for their assistance with the statistical analysis. The authors acknowledge the Siriraj Institute of Clinical Research (SICRES) and Dr. James Mark Simmerman for their efforts to improve the manuscript.

### Funding

The authors received no funding for this work.

### Competing Interests

The authors declare there are no competing interests.

### Author Contributions

- Somthida Vachirapuranon conceived and designed the experiments, performed the experiments, analyzed the data, prepared figures and/or tables, authored or reviewed drafts of the article, and approved the final draft.
- Chodchanok Vijarnsorn conceived and designed the experiments, performed the experiments, analyzed the data, prepared figures and/or tables, authored or reviewed drafts of the article, and approved the final draft.
- Supaluck Kanjanauthai conceived and designed the experiments, authored or reviewed drafts of the article, and approved the final draft.
- Teerapong Tocharoenchok conceived and designed the experiments, authored or reviewed drafts of the article, and approved the final draft.
- Krivikrom Durongpisitkul conceived and designed the experiments, authored or reviewed drafts of the article, and approved the final draft.
- Prakul Chanthong conceived and designed the experiments, authored or reviewed drafts of the article, and approved the final draft.
- Paweena Chungsomprasong conceived and designed the experiments, authored or reviewed drafts of the article, and approved the final draft.
- Thita Pacharapakornpong conceived and designed the experiments, authored or reviewed drafts of the article, and approved the final draft.
- Jarupim Soongswang conceived and designed the experiments, authored or reviewed drafts of the article, and approved the final draft.
- Supattra Rungmaitree conceived and designed the experiments, authored or reviewed drafts of the article, and approved the final draft.
- Charn Peerananrangsee performed the experiments, analyzed the data, authored or reviewed drafts of the article, and approved the final draft.

- Ekarat Nitiyarom conceived and designed the experiments, authored or reviewed drafts of the article, and approved the final draft.
- Kriangkrai Tantiwongkosri conceived and designed the experiments, authored or reviewed drafts of the article, and approved the final draft.
- Thaworn Subtaweesin conceived and designed the experiments, authored or reviewed drafts of the article, and approved the final draft.
- Amornrat Phachiyanukul conceived and designed the experiments, authored or reviewed drafts of the article, and approved the final draft.

## Human Ethics

The following information was supplied relating to ethical approvals (i.e., approving body and any reference numbers):

Siriraj Institutional Review Board, Faculty of Medicine, Siriraj Hospital, Mahidol University [Study number 258/2563(IRB4), COA Si 275/2020].

## Clinical Trial Ethics

The following information was supplied relating to ethical approvals (i.e., approving body and any reference numbers):

Siriraj Institutional Review Board, Faculty of Medicine, Siriraj Hospital, Mahidol University [Study number 258/2563(IRB4), COA Si 275/2020].

## Data Availability

The raw data are available in the Supplementary File.

## Clinical Trial Registration

The following information was supplied regarding Clinical Trial registration:

TCTR20200420003.

## Supplemental Information

Supplemental information for this article can be found online at http://dx.doi.org/10.7717/peerj.14279#supplemental-information.

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
