# Peer review of "Major infections following pediatric cardiac surgery pre- and post-CLABSI bundle implementation"

_PeerJ, doi:10.7717/peerj.14279_

## Round 0.1 · original submission · Minor Revisions

Two experts in this field assessed your manuscript and found the data interesting and worth publication in this journal, after addressing the concerns raised. The main criticisms are related to the study justification, statistical analyses applied to the data, and the manuscript format.

Reviewer 1 ·

Basic reporting

1. The paragraph in lines 77-107 is too long. A paragraph should be a series of sentences that are organized and coherent and are all related to a single topic. The paragraph could be broken down into a few paragraphs, with each explaining the local problem, QI intervention, and QI implementation for example. Furthermore, please try to make the description from line 86 to line 101 more concise.

2. The word "proportion" or "incident proportion" is inappropriately used many times where "incidence" should be used. For example, lines 32, 43, 173, 178, 217, 224, 233, 270, and figure 2.

3. The discussion should focus on important matters based on results. For example, the sentence from lines 204-205 is not supported by the preceding sentences in the same paragraph. The paragraph from lines 206 to 214 simply repeats the results and methods and does not include any meaningful discussion. Furthermore, while I understand various definitions of VAP could affect the incidence rate, explanations for the potential overestimation of VAP incidence in the current study could be more concise. I would be more curious to know why VAP would decrease with the CLABSI bundle. Please be careful not misleading readers in lines 235-236 because VAP incidence reduction was not statistically significant. Lastly, the paragraph in lines 239-250 just cites past reports and lacks any meaningful discussion.

4. I don't think discussions from line 259 to line 263 are necessary. It seems unimportant as a limitation.

Experimental design

1. Line 109-110: The rationale that the CLABSI bundle would reduce VAP and SSI is not explained. Therefore, it is hard to understand this hypothesis. Although this manuscript may not be a QI paper in a strict sense, a QI paper should provide any reasons or assumptions that were used to develop the intervention(s), and reasons why the intervention(s) was expected to work. Refer to SQUIRE 2.0 guidelines for details although I do not expect the authors completely follow the guidelines.

2 .How (With what factors) were the ORs adjusted?

3. Since this is a retrospective before-and-after QI study, it is important to examine adherence to the CLABSI bundle. Without the adherence information, we never know the improved outcomes were the results of CLABSI bundle implementation. If the authors do not have adherence data, it is a major limitation of the study and should be discussed as such.

4. Table 2: What is the reason for reporting the incidence of major infection instead of each infection? If there is a rationale that the CLABSI bundle would be effective in decreasing all the major infections, reporting only the incidence of major infections is acceptable. But I feel the CLABSI bundle would be much more effective in reducing BSI than in reducing VAP or SSI. Pneumonia should be replaced with VAP.

Validity of the findings

1. The conclusion should be drawn based on the results. BSI indeed decreased after the CLABSI prevention bundle implementation. However, I don't think the authors have proven the causation; we don't know if it is the CLABSI prevention bundle that decreased BSI. Furthermore, it is not appropriate to conclude that VAP was reduced because its reduction was not statistically significant. The sentence in lines 270-272 is not an appropriate conclusion either; this is not even discussed in detail before.

Additional comments

Minor comments:
1. lines 77-78: I think this sentence is better positioned in the above paragraph. Because the main topic in the paragraph starting from line 78 is local problems including HAI with CRE.
2. line 133 and lines 163-164: "major infection including BSI, SSI, and VAP." This expression means the major infection can include other types of infection. Please define "major infection" clearly.
3. line 173: 17.6%
4. lines 172-174: I think it would be more appropriate and clear to state that "the incidence of VAP did not decrease significantly".
5. line 199: Again, VAP incidence reduction was not significant. The current expression is misleading.
6. line 252: Please entertain the possible effect of changes in perioperative prophylactic antibiotic strategies on the BSI rate. If appropriate, please explain perioperative prophylactic antibiotic strategies in the method or elsewhere.

Reviewer 2 ·

Basic reporting

The English used throughout the manuscript was clear and unambiguous. Nevertheless, I would like to suggest minor changes throughout the manuscript.
- Please kindly change to "bloodstream" infection throughout the manuscript. eg Line 26-27, 33
- Please countercheck the consistency of the hyphen in the words such as "ventilator-associated pneumonia" or "central line-associated bloodstream infection" throughout the manuscript. eg. Line 33-34.
The literature references, structure, figures, and tables were adequate.
- Line 174 "17.6%"
- Line 253: Indentation is required

Experimental design

The research questions were well-defined, relevant, and meaningful. Several concerns were required to be addressed.
1. I suggest creating another section in the methodology describing the bundles and changes rather than crowding the introduction.
2. Line 86-98: It would be clearer and helpful to the readers to further describe the maintenance procedure of the CLABSI bundle such as the flushing technique, dressing protocol, and thrombosis prevention protocol. Since these factors might also be associated with the risk of bloodstream infection (BSI, CRBSI, CLABSI).
3. Line 99-100: please clarify 3.4: the authors mentioned that bathing was performed postoperatively until the hemoculture was negative. Does this imply that the bathing was done only for patients with positive hemoculture? 3.3 and 3.4 were rather confusing in terms of differences.
4. It was also unclear to me whether the CRE prevention bundle described in the study was a part of the CLABSI bundle or another separate bundle. If it was a separate or additional bundle, please describe when this was started and how the authors outline the possible effect of decreasing infection from which of the bundles. This should be addressed in the discussion part too.
5. In the bundle section, please further describe the training period and the training methods of the staff and faculty members.

Validity of the findings

All underlying data were robust. Nevertheless, I have certain concerns in terms of the statistical analyses of the data.
1. I would like the authors to provide the rationale for choosing the factors to perform the multivariate analyses. Some statistically significant factors from the univariate analyses were not included in the multivariate logistic regression, whilst some were included. For instance, bw < 5 kg and genetic syndrome. The factors such as asplenia, pre-clabsi bundle, cpb time were included despite non-statistically significant outcomes.
2. Table 2 demonstrated that the proportions of the major infections were not statistically different, reflecting the crude OR in Table 3. This should be clarified on why this was statistically significant after multivariate analyses, perhaps in the discussion section.
3. Another additional table outlining the proportion of patients with/without major infections would make a result section clearer and more robust.
4. Is it possible for the authors to clarify the reason for using age < 6 months as a stratification? Since the bathing bundles were performed in children > 2 months old. Would the results be different with the alteration of stratification?
5. A subgroup analysis of bloodstream infection (BSI) alone rather than all major infections might strengthen the effectiveness of the bundles.
6. Conclusion
"The implementation of the CLABSI prevention bundle in our pediatric CCS unit resulted
in important reductions in the incidence proportion of BSI and VAP." I think by including VAP in this statement might be slightly overrated. Please consider certain changes.
7. Does the authors observed any skin irritation effect from the implementation of bundles of chlorhexidine bathing?

Additional comments

Thank you for inviting me to review this manuscript. I wished the authors best of luck in their work.

---

## Round 0.2 · accepted · Accept

After revising the rebuttal letter and the revised version of the manuscript I think all the comments were properly addressed and as a consequence, the manuscript does not need to go out for the second round of peer review and is suitable for publication in Peer Journal.